# Membranous Nephropathy

**DOI:** 10.3390/jcm14030761

**Published:** 2025-01-24

**Authors:** Claudio Ponticelli

**Affiliations:** Independent Researcher, 20131 Milano, Italy; ponticelli.claudio@gmail.com

**Keywords:** membranous nephropathy, glomerulonephritis, podocyte, nephrotic syndrome, proteinuria

## Abstract

Membranous nephropathy is a glomerular disease that may be caused by exogenous risk factors in genetically predisposed individuals (primary MN) or may be associated with other autoimmune diseases, drug exposure, or cytotoxic agents (secondary MN). Primary membranous nephropathy (PMN) is an autoimmune disease in which antigens—mainly the phospholipase A2 receptor—are located in the podocytes and are targeted by circulating antibodies, leading to in situ formation of immune complexes that activate the complement system. Clinically, the disease is characterized by nephrotic syndrome (NS) and associated complications. The outcome of PMN can vary, but untreated patients with NS may progress to end-stage kidney disease (ESKD) in 35–40% of cases within 10 years. Treatment primarily aims to prevent NS complications and progression to ESKD. The most commonly used immunosuppressive drugs are rituximab, corticosteroids, cyclophosphamide, and calcineurin inhibitors. Most patients may experience an improvement of proteinuria, which can sometimes be followed by NS relapse. Fewer than 50% of patients with PMN achieve complete and stable remission. In addition to immunosuppressive therapy, antiproteinuric, anti-lipemic, and anticoagulant medicaments are often required.

## 1. Introduction

Membranous nephropathy (MN) is a glomerular disease clinically characterized by proteinuria (usually in a nephrotic range), edema, and slow progression to end-stage kidney disease (ESKD). Histologically, there is uniform thickening of the glomerular basement membrane (GBM) due to subepithelial deposits of immune complexes, which appear as granular deposits of IgG (primarily IgG4) and C3 on immunofluorescence and as subepithelial electron-dense deposits on electron microscopy.

## 2. Etiology of MN

Membranous nephropathy is traditionally divided into primary MN (PMN) and secondary MN. Little is known about the factors that can trigger the production of antibodies directed against podocyte enzymes or other molecules in PMN. It is likely that exogenous factors may contribute to the conformational changes and increased expression of podocyte antigens. A role of viral or bacterial infection has been suggested [1,2]. Pollution also plays an important role. Several studies outlined the direct association of MN with pollutants, such as fine particulate matter (PM, PM_2.5_–PM_10_) and/or nitrogen oxide [3,4,5]. It has been estimated that each 10 μg/m^3^ increase in PM_2.5_ concentration is associated with 14% higher odds of MN in regions with PM _2.5_ above 70 μg/m^3^ [4]. Genetic predisposition may also be involved in the etiopathogenesis of PMN [6,7,8,9,10]. A large genome-wide association study (GWAS) for PMN was conducted in 3782 cases and 9038 controls of East Asian and European ancestries. This GWAS discovered an unusual genetic architecture of MN, with two non-HLA loci that are transcriptional regulators of inflammation—Nuclear factor kappa B subunit 1 (NFKB-1) and Interferon-regulator factor 4 (IRF4)—and two HLA alleles—either DRB1*1501 in East Asians (OR = 3.81) or DQA1*0501 in Europeans (OR = 2.88) and DRB1*0301 in both ethnicities (OR = 3.50). GWAS loci explained 32% of MN risk in East Asians and 25% of the risk in Europeans and showed that a single haplotype at the PLA2R1 locus carries the disease risk in both East Asians and Europeans and exhibits genetic interactions with HLA-DRB1 risk alleles [11].

Thus, it is possible that PMN is triggered by undiagnosed viral infections or pollutants, which in genetically predisposed individuals can induce exposure to podocyte antigens, triggering autoimmune responses.

Secondary MN may be associated with bacterial or viral infections, including hepatitis B, hepatitis C, human immunodeficiency virus, varicella zoster, COVID-19 infection, and vaccination [12,13,14,15,16]. Other cases are frequently associated with other autoimmune diseases, such as systemic lupus erythematosus and rheumatoid arthritis [17,18], or malignancy, especially carcinoma [19,20,21]. How extensive the investigation for detecting an ‘occult’ underlying cancer in a patient with MN should be is still a matter of debate. While current guidelines prioritize the measurement of serum anti-PLA2R antibodies for diagnosing primary MN, it has been reported that several secondary MN cases may also test positive for serum anti-PLA2R (see later). At least in heavy smokers and in elderly patients, however, an extensive investigation is advisable. In these patients and in those with circulating anti-thrombospondin 1 domain containing 7 A antibodies, in addition to a thorough history and physical examination, a chest radiogram (preferably a computed tomography because of its higher sensitivity), a digital rectal exam of the prostate, a thorough breast examination and mammography in women, and renal ultrasonography (usually already performed in connection with a kidney biopsy) are suggested. A gastroscopy and a colonoscopy (or at the very least a stool occult blood examination) can also be performed in older patients or in the presence of upper gastrointestinal symptoms, early satiety, unexplained weight loss, or iron deficiency anemia. MN may also occur after exposure to drugs (lithium, non-steroidal anti-inflammatory drugs, clopidogrel, probenecid, etc.) or cytotoxic agents (formaldehyde, heavy metals, hydrocarbons, etc.).

## 3. Physiopathology of PMN

While the etiology of PMN remains elusive, experimental and clinical studies have clarified its pathogenesis. In 1959, Heymann et al. first demonstrated in rats that a disease similar to the human MN can be caused by an in situ deposit of immune complexes resulting from the reaction of a circulating antibody against an antigen planted in the subepithelial position [22]. Heymann nephritis can be subdivided into active and passive nephritis models. Active nephritis may be induced by immunizing specific strains of rats with fractions of proximal tubular brush border; passive nephritis is induced by a single injection of anti-brush border anti-serum (anti-Fx1A). Megalin, a 330-molecular-weight glycoprotein located in the brush border of the proximal convoluted tubule and in the epithelial cell membrane of the glomerulus, was identified as the possible antigen involved in Heymann nephritis [23,24]. However, megalin is not involved in human MN. In a pivotal study, Salant et al. demonstrated that proteinuria in the passive model of MN is complement-dependent [25]. Depletion of C6, which is part of the cytolytic C5b-C9 complex, prevented the development of proteinuria. In addition, the activation of the C5b-C9 complex in podocytes depends on C6 contribution [26,27,28,29,30]. C5b-9, also called membrane attack complex (MAC), can stimulate podocytes to produce a number of inflammatory mediators, including transforming growth factor (TGF)-beta and reactive oxygen species (ROS), that may impair the glomerular filtration barrier by peroxidation of membrane proteins and collagen. MAC-activated cells have also been found to express proinflammatory proteins, often through NFKB-dependent transcription, assemble inflammasomes, and facilitate secretion of interleukin (IL) IL-1β and IL-18, as well as other signaling pathways [31].

The studies based on Heymann models stimulated several speculations that may also be applied to human MN, but the search for the potential antigen (s) remained elusive for many years. In 2002, a seminal study demonstrated that antibodies against a podocyte antigen can induce MN in humans through mechanisms similar to those proposed for Heymann nephritis. Debiec et al. reported a case of MN in a newborn born by a neutral endoproteinase (NEP)-deficient mother. The Authors found that the disease was mediated by maternal anti-NEP antibodies that formed immune complexes with NEP on the podocyte membranes of the infant [32]. A few years later, Beck et al. [33] identified a second antigen in adults. They found that up to 70% of adults with PMN exhibited circulating IgG4 antibodies against the M-type phospholipase receptor (PLA2R) expressed on the podocyte surface. Other reports confirmed the presence of anti-PLA2R antibodies in 65–80% of patients with MN and outlined not only the diagnostic but also the prognostic role of anti-PLA2R antibodies. Indeed, high titers of PLA2R antibodies are associated with decreased rates of response to treatment, while low levels of antibodies are associated with a high rate of spontaneous remission [34,35,36,37,38,39,40]. However, PLA2R is not the only antigen in PMN. In 2–10% of patients with MN, circulating antibodies are directed against thrombospondin 1 domain containing 7 A expressed on the podocyte surface [41,42,43]. These antibodies are pathogenic [44,45] and can often be found in MN associated with cancer [46,47,48]. Novel antigens of primary and secondary MN have been detected by mass spectrometry or immunochemistry: exostosin 1 and 2 in type V lupus nephritis, NELL-1 (often associated with malignancy), semaphorin 3B, protocadherin-7, serine protease AHTRA1, and contactin 1 [49,50,51]. More recently, seven other putative antigens were discovered within immune complexes from biopsies of patients with MN, including seizure-related 6 homolog like 2 (SEZ6L2), vasorin, early endosome antigen 1, macrophage stimulating, natriuretic peptide receptor 3, ficolin 3, and cluster of differentiation 206. Frequencies of each antigen, determined by staining consecutive case series, ranged from under 1 to 4.9% [52]. Thus, an elevated number of antigens may be discovered in MN, most of them being localized on the podocyte surface.

One may wonder why so many podocyte antigens are involved in MN. It is possible that exogenous factors might induce in genetically predisposed individuals a conformational change of the epitopes in some podocyte proteins (more frequently PLA2R). Research on the epitope of PLA2R demonstrated that the main epitope is in an outer cysteine-rich (CysR) domain [53,54]. Two inner epitopes have also been found, respectively, in the C-type lectin domains (CTLDs) 1 and 7. While the disease progresses, the target epitope may change from outer CysR to the inner CTDL1 to CTDL7 [55]. This epitope spreading can be associated with a poor prognosis [56]. Patients with antibodies against the inner epitopes tend to be older and resistant to therapy [57].

Dysregulated autophagy may also play an important role. Autophagy is a cellular degradative pathway that involves the delivery of cytoplasmic cargo to the lysosome for degradation and eventual recycling of cytoplasmic components. In autoimmune diseases, an upregulated autophagy can induce inflammation and activate innate and adaptive immunity [58]. On the other hand, podocytes have a high level of autophagy that can protect them from injury [59]. Experimental studies showed that defective autophagy can increase the susceptibility of podocytes to injury, leading to podocytopenia, foot effacement, and increased expression of podocyte proteins [60]. Secretory group IB of PLA2R (PLA2-IB) can downregulate autophagy and induce podocyte apoptosis, contributing to podocyte injury via PLA2R [61]. In MN patients with nephrotic-range proteinuria, significantly higher levels of secretory PLA2-IB and fewer autophagosomes were seen in comparison with non-nephrotic patients, confirming defective autophagy in podocytes and abnormal expression of podocyte proteins and enzymes [62]. Unmasked “hidden” antigens are recognized as a danger-associated molecular pattern (DAMP) by the pattern recognition receptors (PRRs) of the innate immunity. Autophagy has a critical role in this phase since it may deliver DAMPs to toll-like receptors, NOD-like receptors, and inflammasomes [63,64]. PRRs promote a signal, amplified by adapter molecules (MyD88, Tirap, Trif, Tram) and autophagy proteins (Atg16), which is transmitted by a chain of kinases (IRAK1, IRAK4, TBK1, IKK) to inflammatory transcription factors like nuclear factor kappa B, activator Protein-1, and Interferon-3 that regulate the inflammatory response [65,66]. In the inflammatory microenvironment, dendritic cells become mature and migrate to lymph nodes, where they recognize the intercepted podocyte protein as an antigen and present the antigen to naive T cells (Figure 1). Once the T cell has received the co-stimulation signals of activation, calcineurin, a complex of phosphatases, dephosphorylates a family of proteins called nuclear factor-activating T cell, allowing its entrance into the nucleus, where it participates in the synthesis of IL-2, which binds to its receptor and, together with IL15, activates Janus 3 kinase and delivers growth signals through phosphatidylinositol 3 kinase and protein kinase B. The mechanistic target of rapamycin (mTOR) plays a key regulator role in protein translation. Several cytokines generate different types of T helper cells: Th1 (IL-12), Th17 (IL-17), or Th2 (IL-4). The collaboration between T and B cells activates B cells to produce plasmablasts and antibodies [67].

In patients with PMN, the antibodies are usually IgG4-directed against the podocyte antigen, with in situ formation of immune complexes.

In response to the subepithelial deposition of immune complexes, the complement system is activated and plays a central role in PMN [68]. There are three pathways that can activate the complement cascade: the classical, lectin, and alternative pathways (Figure 2). Usually, antibodies activate the classical complement pathway by binding C1q to the Fc portions of IgG1-3 and IgM. However, in PMN, the dominant antibody is IgG4 [33], which cannot bind C1q [69]. In a cell culture model, sera or IgG4 of anti-PLA2R1-positive patients induced proteolysis of synaptopodin and nephrin 1 in the presence of complement, resulting in abnormalities of the podocyte cytoskeleton. These effects were not observed in IgG4-depleted sera, suggesting that IgG4 can activate the lectin pathway of complement in patients with PMN who are positive for anti-PLA2R antibodies [70]. The alternative pathway may also be involved. This has been indirectly demonstrated by the deterioration of kidney function in MN patients with circulating antibodies against factor H, a regulator of the alternative pathway of complement [71]. Finally, complement was primarily activated via the classical pathway in a murine model of MN [72].

The formation of C3 convertase initiates the generation of C5 convertase.

In the classical and lectin pathways, C5 convertase is formed by the binding of C3b to C4b,2a. In the alternative pathway, C5 convertase is formed by the binding of Bb serum protease to C3 convertase to form C3bbBb. The cleavage of C5 generates the proinflammatory C5a and C5b, which form a cytolytic complex with C6,7,8,9 (C5b–C9), also called membrane attack complex (MAC).

Thus, in PMN, the cascade of complement may be activated by multiple pathways, eventually leading to the formation of C5 convertase, which splits C5 into C5a and C5b-9, also called MAC. As a result of the cytolytic effects and of reactive oxygen species at the GBM, MAC may promote podocyte effacement, disruption of the actin cytoskeleton, and proteinuria [73,74]. Apart from MAC, elevated levels of anaphylatoxin C3a and C5a have been detected in podocytes of MN [75,76]. Antagonists of the C3a receptor could prevent the oxidative stress produced by the serum of humans with PLA2R-positive MN [77]. The final result of immune complex deposits on GBM and complement activation is proteinuria, often in the nephrotic range (>3.5 g per day).

## 4. Physiopathology of the Nephrotic Syndrome

The nephrotic syndrome (NS) is usually defined by the presence of proteinuria exceeding 3.5 g/24 h, hypoalbuminemia, hypercholesterolemia, and various degrees of edema. It may complicate many kidney diseases and is frequently associated with MN. Much attention has been paid to the possible progression of MN. However, the complications of NS should not be neglected since they can cause severe consequences, particularly in patients with persistent NS.

*Edema:* is a constant sign of NS. It may range from a mild pitting edema to a true anasarca. According to the current hypothesis, edema is a consequence of sodium retention caused by an increased avidity to NaCl of collector tubules, leading to an expansion of vascular volume. The sodium retention is probably due to the presence of plasminogen in the filtered proteins. Plasminogen may be converted to plasmin (a protease) with activation of the epithelial sodium channel in the collecting ducts [78]. Other proteases, such as cathepsin B [79], can contribute to sodium retention in NS.

Swollen areas are at increased risk of skin ulcers and infection. Edema can also impair blood circulation, leading to intravenous coagulation and vein thrombosis, particularly after long-term sitting. In severe cases of NS, anasarca, dyspnea, and congestive heart failure may develop.

The Cure Glomerulonephropathy study reported that edema had the strongest association with health-related quality of life (HRQOL) in patients with primary glomerular diseases, with substantially greater impact than other demographic and clinical variables [80].

*Proteinuria:* Albumin and other proteins that pass the glomerular barrier are reabsorbed into proximal tubule cells under the government of two glycoprotein endocytic receptors: cubilin and megalin [81]. The reabsorption of this unwanted material activates macroautophagy. Albumin is sequestered in a double-membrane structure (phagophore) that is elongated and closed to form vesicles (autophagosomes) that fuse with lysosomes to form autolysosomes [82]. Lysosomal hydrolytic enzymes completely degrade the albumin into macromolecules that are poured for transcytosis into the peritubular capillaries [83]. Excessive reabsorption of proteins inhibits both proximal tubular uptake and degradation capacity, resulting in proteinuria and tubular damage [84]. In addition, injured tubular cells release into the interstitial space a lot of proinflammatory cytokines, cells, and transcription factors that mediate the development of tubular atrophy and interstitial fibrosis, independent of the type of underlying glomerulopathy [85,86,87,88,89].

Several mechanisms can dysregulate autophagy. Excess protein intake in proximal tubular cells may impair autophagy and increase apoptosis, leading to tubular atrophy [90]; in turn, damaged tubular cells release endothelin, cytokines, and inflammatory transcription factors that can promote epithelial–mesenchymal transition and fibrosis [91,92]. Autophagy plays an essential role in maintaining renal tubular cell integrity during stress conditions by eliminating damaged mitochondria and avoiding the accumulation of aggregate-prone proteins [93]. Defective autophagy in tubular epithelial cells can increase the susceptibility of renal cells to injuries and favor the production of TGF-β1, which can exert different functions by promoting fibrosis and interstitial fibrosis or inducing protective autophagy [94]. However, TGF-β1 may also collaborate with C3a and the C3a receptor in promoting epithelial–mesenchymal transition [95]. The final result is the development of chronic kidney disease, which can slowly progress to ESKD in untreated patients [96,97].

### 4.1. Hypoalbuminemia

Hypoalbuminemia is part of the definition of NS. Loss of albumin in urine and increased tubular degradation may partially explain the low levels of serum albumin in patients with NS. In addition, kidney inflammation increases capillary permeability and escape of serum albumin into the interstitial space, leading to its expansion and increasing the distribution volume of albumin while shortening the half-life and decreasing total albumin mass [98].

### 4.2. Dyslipidemia

Dyslipidemia is a frequent complication of MN, particularly in patients with nephrotic syndrome. Different factors contribute to the development of lipid disorders in NS. The production of low-density lipoproteins (LDLs) and very low-density lipoproteins (VLDLs) is increased, while enzymes involved in the catabolism of cholesterol and triglycerides (TG) are lost in the urine [99].

Hypoalbuminemia and reduced oncotic pressure increase the transcription and synthesis of proteins produced by the liver, including apoB, apoE, apoCII, lipoprotein-a, 3-hydroxy-3-methylglutaryl-Coenzyme A (HMG-CoA) reductase, angiopoietin-like proteins 3 and 4, and the enzyme Proprotein Convertase Subtilisin/Kexin type 9 (PCSK9). These proteins can induce elevations in LDL and VLDL cholesterol. Normally, cholesterol contained in VLDLs and intermediate-density lipoproteins (IDLs) is transferred to LDLs and then removed from the circulation by LDLs binding to specific receptors in the liver and, to a lesser extent, in extrahepatic tissues. However, in NS, the upregulation of PCSK9 [100] degrades LDL receptors, thus inhibiting cholesterol’s cellular uptake and increasing plasma cholesterol [101]. On the other hand, TGs contained in VLDLs interact with lipoprotein lipases in the circulation to form IDLs, which are metabolized into LDLs, while their TGs are hydrolyzed into free fatty acids by hepatic lipases [102].

An important role in the genesis of dyslipidemia in NS is also played by the urinary loss of small-sized high-density lipoprotein (HDL) particles and factors such as apolipoprotein C-II, an activator of LPL, and lecithin-cholesterol acyl transferase, an enzyme that esterifies cholesterol and raises HDL components [103]. Moreover, the loss of HDL decreases the binding of HDL with the anti-inflammatory lipoprotein apoM (produced by renal tubular epithelium) [104].

The importance of high LDL cholesterol as a risk factor for cardiovascular disease is well established in the general population. Although a correlation between dyslipidemia and mortality is not well defined in patients with NS, the Kaiser study reported that patients with primary NS had a higher risk of cardiovascular outcomes and death [105]. Evidence suggesting that renal lipid accumulation may lead to kidney dysfunction has mounted over recent years. Lipid accumulation, changes in circulating adipokines, alterations in renal lipid metabolism, mitochondrial dysfunction, generation of reactive oxygen species, and endoplasmic reticulum stress eventually lead to alterations in the glomerular filtration barrier, podocyte injury, and kidney failure [106,107]. It has also been demonstrated that the accumulation of lipids in the form of lipid droplets in podocytes may lead to mitochondrial dysfunction and inflammation [108]. Dyslipidemia may be involved in tubulo-interstitial fibrosis. In NS, lipoproteins and lipid droplets are part of the tubulo-toxic proteins abnormally filtered by the damaged GBM. Abnormal production of free fatty acids (FFAs) can further aggravate the injury in renal proximal tubular cells. Those cells use ATP, mostly derived from the oxidation of fatty acids, to obtain energy. Under normal conditions, serum albumin binds to fatty acids, keeping the levels of FFAs low. In NS, the low levels of serum albumin prevent FFAs from binding albumin; thus, in tubular cells of nephrotic patients, there is an accumulation of FFAs, leading to dysregulation of mitochondrial oxidation, which can reduce energy production and promote epithelial-to-mesenchymal transition, inflammation, and interstitial fibrosis [109]. The tubulo-interstitial damage can eventually result in chronic kidney disease, outlining the relevance of persistent lipid disorders in patients with NS.

### 4.3. Hypercoagulable State

Nephrotic proteinuria can be associated with urinary loss of several regulators of coagulation, leading to insufficient activity of plasminogen activator inhibitor-1 (PAI-1) [110], reduced fibrinolysis [111], and antithrombin defects [112]. On the other hand, hypoalbuminemia and dyslipidemia may promote an increase in procoagulant factors [113], platelet hyperreactivity [114], and plasma viscosity [115]. Angiotensin II, via angiotensin receptor type 1, can promote thrombosis by accelerating fibrin formation and increasing plasma levels of PAI-1 [116]. Finally, an analysis of the current literature concluded that the combination of secretory PLA2 may promote the release of arachidonic acid from membrane phospholipids and induce platelet aggregation in MN with circulating anti-PLA2R antibodies [117].

The combination of these factors can result in an increased risk of thrombosis in patients with PMN and NS. It has been estimated that arterial thrombosis is eight times more frequent in patients with PMN than in age- and sex-matched populations [118]. Renal vein thrombosis is a rare condition, but it is relatively frequent in patients with PMN. In a retrospective review of the Toronto register of patients with idiopathic glomerular diseases, the risk for venous thromboembolic events was highest in patients with MN, with an adjusted hazard ratio of 10.8 [119]. It has been reported that circulating anti-alpha-enolase antibodies are present in 70% of patients with MN [120]. Since alpha-enolase is a plasminogen-binding protein [121], its inhibition by specific antibodies might explain the defective fibrinolysis in MN.

## 5. The Natural Course of PMN

The outcome of PMN is variable. For a long period, the natural course of the disease was stated to follow the rule of thirds, with about one-third of patients entering spontaneous remission, one-third showing persistent proteinuria, and one-third progressing to ESKD. However, if one considers the natural course of PMN with NS 10 years after clinical onset, it appears that 35–40% require regular renal replacement therapy [95,96]. On the other hand, about 40% of patients with PMN do not develop NS and have a better prognosis [122]. Thus, the rule of thirds may be ignored, and the outcome of patients may be stratified into three groups with different prevalence: (a) Low-risk group (normal renal function and less than 4 g/day proteinuria over a 6-month period). These patients have a less than 5% risk for progression over a 5-year period. (b) Medium-risk group (normal renal function and persistent proteinuria of between 4 and 8 g/day over a 6-month period). This group has a high risk of poor outcomes without treatment. (c) High-risk group (worsening renal function and proteinuria greater than 8 g/day over a 6-month period), with the highest risk of ESKD [123].

## 6. Treatment

Management of MN involves immunosuppressive treatment aimed at preventing the deleterious effects of autoantibodies leading to progressive kidney disease and symptomatic treatment aimed at preventing the toxicity of extrarenal complications of NS.

### 6.1. Immunosuppressive Treatment

According to the KDIGO guidelines [124], immunosuppressive treatment should be reserved for patients with at least one factor for disease progression (NS, reduced GFR, thromboembolic events, or acute kidney injury). In these cases, rituximab or cyclophosphamide and glucocorticoid for 6 months or tacrolimus for 6 months, depending on the estimated risk, are suggested.

Historically, a multicenter RCT in 1984 reported that cyclical therapy based on alternating corticosteroids with an alkylating agent (chlorambucil) every other month for 6 months significantly increased the probability of NS remission and protected kidney function compared with symptomatic therapy [125]. These results were confirmed by two other RCTs with the same therapeutic regimen [126,127]. Another Italian RCT compared the efficacy and safety of 6-month regimens of alternating corticosteroids with either chlorambucil (0.2 mg/kg/day) or cyclophosphamide (2 mg/Kg/day). Patients on cyclophosphamide achieved a higher number of remissions and a lower number of side effects than those assigned to chlorambucil [128]. In 2007, an Indian RCT compared the results of a 6-month cyclical therapy with cyclophosphamide alternated with steroids vs. symptomatic treatment. After a mean follow-up of 11 years, 72% of treated patients entered complete or partial remission vs. 35% of the controls. At 10 years, 89% of treated patients and 65% of controls were alive without dialysis. Serious infections occurred in 15% of treated participants vs. 24% of controls, and no case of malignancy was reported [97].

Observational studies reported that cyclosporine or tacrolimus may reduce proteinuria. A review of 918 patients with PMN treated with either cyclosporine or tacrolimus showed that 259 (28.2%) entered complete remission and 372 (40.5%) achieved partial remission of proteinuria; however, relapse was frequent after drug reduction or withdrawal [129]. A Spanish RCT assigned 48 patients with PMN to receive tacrolimus (0.05 mg/kg/day) over 12 months, with a 6-month taper (25 patients) or symptomatic therapy (23 patients). At 18 months, the probability of remission (either complete or partial) was 94% in the treatment group vs. 35% in the control group. The decrease in proteinuria was significantly greater in the treatment group. However, NS reappeared in almost half of the patients who were in remission by the 18th month after tacrolimus withdrawal [130]. This writer does not believe in trough levels. Some investigators suggest monitoring calcineurin inhibitors (CNIs) based on trough levels; however, while therapeutic drug monitoring can provide information on serum concentrations, it does not offer insights into intracellular levels. Additionally, various conditions and medications can interfere with cytochrome enzymes and the efflux transporter P-glycoprotein. This may lead to serious errors since elevated CNI levels in the blood may be associated with low intracellular concentrations and vice versa.

A review of the observational tri-efflux transporter P-glycoprotein also reported that of 353 patients with PMN who were treated with rituximab, 102 (28.8%) entered complete remission of proteinuria,147 (41.6%) achieved partial remission, and 104 (29.6%) did not respond [131]. Four randomized controlled trials (RCTs) have been conducted. The first multicenter RCT randomized 75 patients with PMN to renin–angiotensin system inhibitors (RASis) plus rituximab—375 mg/m^2^ on days 1 and 8—or RASis alone. At 6 months, 13 (35%) patients given RASis and rituximab and 8 (21%) controls achieved complete or partial remission. In a post-hoc analysis after a mean period of 23 months, the remission rate increased to 24/77 patients (65%) in the rituximab group vs. 13/38 (34%) in controls. Complete remissions were, respectively, 19% vs. 2.6% [132]. Another multicenter RCT assigned 130 patients with PMN to rituximab—1 g × 2 repeated after 6 months—or cyclosporine—3.5 mg/kg/day for 12 months. The cumulative incidence of complete plus partial remission at one year was not different between the two groups (60% vs. 52%). At 24 months, the number of those in remission remained stable in the rituximab arm (60%, of which 35% were in complete remission), while it fell to 20% in the cyclosporine group due to relapses after drug discontinuation [133]. Considering the quality-adjusted life years, rituximab may be a cost-effective option for the treatment of membranous nephropathy when compared with cyclosporine [134].

The STARMEN RCT compared, in 86 patients with PMN, the efficacy of sequential therapy of tacrolimus (full dose for six months and tapering for another three months) plus rituximab (one gram at month six) versus six-month cyclical treatment with corticosteroid and cyclophosphamide. At 24 months, 11 patients (26%) in the tacrolimus–rituximab group and 26 patients (60%) in the corticosteroid–cyclophosphamide (cyclical therapy) group were in complete remission. Complete or partial remission occurred in 25 patients (58.1%) in the tacrolimus–rituximab group and in 36 patients (83.7%) in the corticosteroid–cyclophosphamide group. Serious adverse events developed in three patients undergoing tacrolimus–rituximab therapy and one patient undergoing the cyclical therapy. Also, relapses occurred in three patients in the tacrolimus–rituximab group and one patient in the cyclical therapy group. Anti-PLA2R titers showed a significant decrease in both groups, but the proportion of anti-PLA2R-positive patients who achieved depletion of anti-PLA2R antibodies was significantly higher at six months in the corticosteroid–cyclophosphamide group (92%) compared to the tacrolimus–rituximab group (70%) [135].

In the Ri-Cyclo RCT, 74 adults with PMN and NS were randomized to receive rituximab (1 g on days 1 and 15) or a 6-month cyclical therapy with corticosteroids alternated with cyclophosphamide every other month. At 12 months, six of 37 patients (16%) randomized to rituximab and 12 of 37 patients (32%) randomized to the cyclical regimen entered complete remission. At 24 months, the probabilities of complete remission were 42% and 43%, respectively. The probabilities of complete plus partial remission were 83% and 82%, respectively. The safety profiles were similar [136].

A Chinese retrospective study of PMN compared the outcomes of 40 patients treated with rituximab vs. 23 given cyclical therapy with corticosteroids–cyclophosphamide. No differences in efficacy and safety were found between the two regimens at 2 years, but the risk of relapse was significantly more frequent in the rituximab group [137].

In an Indian study, 64 participants with PMN and an eGFR < 60 mL/min/m^2^ were assigned to receive either rituximab or cyclical therapy with corticosteroids and cyclophosphamide. At 24 months, 30 (47%) patients were in remission and eight had progressed to ESRD. Rituximab and cyclical therapy were equally effective, but rituximab had fewer adverse events [138].

Combination therapy with rituximab and short use of cyclophosphamide and prednisone resulted in partial remission in 100% of cases and complete remission in 93% of 15 patients with PMN [139]. In another observational trial, similar treatment with rituximab (1 g on days 1 and 15), cyclophosphamide (2.5 mg/kg/day for 1 week and 1.5 mg/kg/day until week 8), and prednisone (60 mg/day tapered off over 28 weeks) was given to patients with PMN. Among 60 participants, 45% entered complete remission at one year and 88% at 3 years. Four patients relapsed after B cell reconstitution. Fourteen serious adverse events occurred [140]. Anti-drug antibodies (ADAs)may develop in Rituximab-treated patients; however, the long-term implications of ADAs in PMN are poorly understood.

The available data show that cyclical therapy and rituximab may give similar results. Rituximab is easier to handle and is the preferred initial treatment in Western countries; however, the optimal dosages of rituximab for initial and maintenance therapies are still unclear. Also needed is a careful assessment of long-term safety and efficacy outcomes since all the available trials with rituximab have a follow-up of no longer than 24 months [141]. New expensive drugs are under investigation in patients with PMN who are resistant to current therapies. These include new humanized anti-CD20 monoclonal antibodies (obinutuzumab), monoclonal antibodies targeting CD38 expressed on immune cells (Daratumumab), complement inhibitors (Iptacopan and Pegcetacoplan), and chimeric antigen receptor T-cell therapy.

### 6.2. Symptomatic Treatment

Loop diuretics are the first-line treatment in the management of nephrotic edema. They can be used even in patients with depressed kidney function. Oral furosemide, along with modest sodium intake, should be sufficient in case of mild edema. Severe edema requires high-dose oral furosemide (250–500 mg) given in two or three divided doses daily or by continuous intravenous infusion. The combination of furosemide and thiazide can double urine sodium excretion but may induce hypokalemia. Adding a mineralocorticoid receptor antagonist may increase natriuresis and prevent hypokalemia.

Renin–angiotensin system inhibitors are recommended to reduce the amount of proteinuria and its burden. Further support may be given by sparsentan, a dual endothelin and angiotensin II type 1 receptor antagonist [142], or sodium–glucose cotransporter 2 inhibitors [143].

Statins are the first-line antidyslipidemic drugs used in patients with persistent dyslipidemia. Statins decrease cholesterol biosynthesis by inhibiting HMG-CoA reductase (rate-limiting step in the mevalonate pathway) and may induce other beneficial “pleiotropic effects”. Despite the fear of statin-associated myopathy, a systematic review and meta-analysis showed that statins increase relative and absolute risks of myopathy (RR 1.08) but reduce the risk of cardiovascular disease and all-cause mortality (RR 0.74) [144]. Coadministration of statins with ezetimibe, which prevents intestinal cholesterol absorption, elicits additive (−65%) cholesterol lowering. Should statin–ezetimibe fail to reduce cholesterolemia, anti-PCSK9 monoclonal antibodies (alirocumab, evolocumab), bempedoic acid (an inhibitor of ATP citrate lyase), or the siRNA inclisiran may be added [145].

Whether patients with MN should receive anticoagulation therapy to prevent vascular thrombosis still remains to be discussed. According to decision analysis, the benefits of prophylactic administration or oral anticoagulants outweigh the risks in nephrotic patients with PMN [146]. A retrospective analysis of 898 patients with PMN reported that benefit-to-risk ratios increased with worsening hypoalbuminemia from 4.5:1 for albumin under 3 g/dl to 13.1:1 for albumin under 2 g/dl in patients at low risk of bleeding. Patients at intermediate bleeding risk with albumin under 2 g/dl had a benefit-to-risk ratio of 5:1 [147]. The current opinion is that prophylactic anticoagulation should be reserved for patients with MN and high venous thromboembolism risk, i.e., serum albumin < 2 or 2.5 g/dL, morbid obesity, immobilization, or prior history of thrombophilia. Recent papers have reported that the use of direct oral anticoagulants (DOACs) in NS was effective and safer than warfarin [148,149]; nonetheless, it should be considered that the pharmacokinetics of DOACs are influenced by proteinuria and reduced kidney function. DOACs are also susceptible to altered metabolism by P-glycoprotein inhibitors such as calcium channel blockers, calcineurin inhibitors, and glucocorticoids [150].

## 7. Conclusions

Tremendous progress has been made in understanding the pathophysiology and management of MN; however, several aspects remain to be clarified.

Most studies have been conducted in Western countries, leading to the conclusion that the incidence of MN is low, approximately 1/100,000, of which approximately 80% are PMN. However, these data do not account for populations in underdeveloped countries (the vast majority) who are particularly exposed to infections, counterfeit drugs, severe pollution, and other potential causes of MN.

Advanced techniques, such as glomerular microdissection laser and mass spectrometry, have led to the discovery of numerous podocyte antigens. Why so many podocyte antigens potentially play a role in the pathogenesis of MN remains unclear. Little is known about the potential role of T cells and /or dysregulated autophagy in triggering autoimmune responses and compromising podocyte integrity and function.

Despite the use of various immunosuppressive drugs and monoclonal antibodies, many patients with PMN either do not respond, cannot tolerate prescribed medicaments, or experience frequent relapses necessitating further treatments. Emerging therapies include new anti-CD20 monoclonal antibodies such as obinutuxumab, monoclonal antibodies targeting CD38-positive plasma cells, C3 blockers, and other complement inhibitors (C5aR, C5b-C9, C3aR, etc.).

All these potential advancements and proposed therapies are extremely costly. What is urgently needed is an effective and safe treatment that is also accessible to low-resource countries.

## Figures and Tables

**Figure 1 jcm-14-00761-f001:**
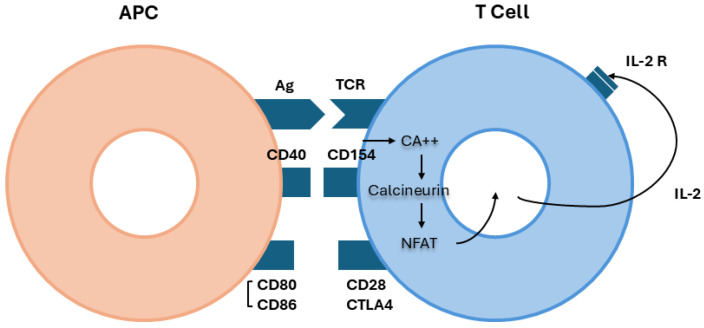
Activation of T cells. The contact between the antigen-presenting cell (APC) and T cell receptor (TCR) (signal 1) is an “anergic-apoptotic” signal that is inadequate for activating T cells. A second signal (co-stimulation) is needed. Co-stimulation requires contact between adhesion molecules on the surface of APCs (CD40, CD 80, CD 86) and molecules on the surface of T cells (CD28). Instead, the cytotoxic T lymphocyte antigen 4 (CTLA4) can inhibit the signal 2 of T cell activation. After co-stimulation, there is a large influx of calcium ions into the cytoplasm that activates a system of phosphatases called calcineurin. This dephosphorylates a family of proteins called nuclear factor-activating T cell (NFAT), thus allowing its entrance into the nucleus where NFAT participates in the synthesis of interleukin-2 (IL-2).

**Figure 2 jcm-14-00761-f002:**
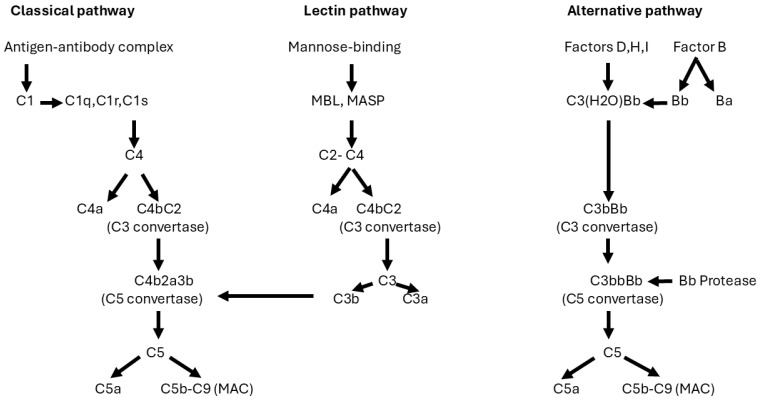
The complement cascade. Complement can be activated through three pathways: classical, lectin, and alternative. The classical pathway is triggered by antibody (IgG 1–3)–antigen complexes binding to C1, which has three subcomponents: C1q, C1r, and C1s. The pathway forms a C3 convertase, C4b2a, which splits C3 into two fragments: C3a (anaphylatoxin), which can promote inflammation, and C3b, which can attach to and opsonize the pathogens. The lectin pathway is activated by the binding of mannose-binding lectin (MBL) to mannose residues on the pathogen surface. The mannose binding activates the MBL-associated serine proteases, MASP-1 and MASP-2, which activate C4 and C2 to form the C3 convertase, C4b2a. The alternative pathway is initiated by the spontaneous hydrolysis of C3. The cleavage of factors B, D, H, and I induces a conformational change in C3 (OH), which interacts with C3b to form a C3 convertase, C3bBb.

## Data Availability

The data supporting the findings of this study are available within the article.

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
