# Peer review of "Membranous Nephropathy"

_jcm, 2025, doi:10.3390/jcm14030761_

Round 1

Reviewer 1 Report

Comments and Suggestions for Authors

This is a comprehensive summary of the pathophysiology and current management strategies for the treatment of Membranous nephropathy.  It serves as a supplementary guide for membranous that might be useful for trainees or nephrologists looking to review this subject area.  I would support the publication with minor revisions. 

I have a few suggestions to improve the paper:

For the section on edema, the authors can cite this article from the CURE-GN study which demonstrates that edema consistently has a the largest impact on the quality of life of patients with glomerular disease. https://pubmed.ncbi.nlm.nih.gov/30898342/

Regarding the treatment of Membranous, it may be helpful to mention this economic evaluation of the MENTOR trial which demonstrated that Rituximab may be a cost effective strategy compared to cyclosporine when considering the Quality Adjusted Life Year benefit. https://pubmed.ncbi.nlm.nih.gov/38621719/

Author Response

Comment 1: For the section on edema, the authors can cite this article from the CURE-GN study which demonstrates that edema consistently has a the largest impact on the quality of life of patients with glomerular disease. https://pubmed.ncbi.nlm.nih.gov/30898342/

Response 1: The Cure Glomerulonephropathy study reported that edema had the strongest association with health-related quality of life (HRQOL) in patients with primary glomerular diseases, with substantially greater impact than other demographic and clinical variables [80]

Comment 2: Regarding the treatment of Membranous, it may be helpful to mention this economic evaluation of the MENTOR trial which demonstrated that Rituximab may be a cost effective strategy compared to cyclosporine when considering the Quality Adjusted Life Year benefit. https://pubmed.ncbi.nlm.nih.gov/38621719/

Response 2: Considering  the quality adjusted life year, rituximab may be a cost-effective option for the treatment of membranous nephropathy when compared with cyclosporine [134]

Reviewer 2 Report

Comments and Suggestions for Authors

I read with interest this very clear and comprehensive review about MN pathophysiology and management. Here are my remarks and suggestions:

1. Could you detail which auto-immune diseases (lupus, IgG4-RD, etc.) and which cancers are the most associated with secondary MN?

2. SLGT2 inhibitors should also be mentioned as efficient molecules to decrease proteinuria as symptomatic treatment.

3. Are high anti-PLA2R levels and the presence of epitope spreading in favour of early immunosuppresive treatment?

4. Please discuss the potential appearance of anti-Ritiximab antibodies and the potential interest of Obinutuzumab in patients with relapse of PLA2R-positive PMN after Rituximab displaying anti-RTX antibodies. 

5. In your opinion, should malignancy be looked for in all cases of PMN?

6. Given that MN are at higher risk of thrombosis than nephrotic syndrome from other causes, some teams, notably in France, consider a higher threshold of albuminemia (< 25 g/dL) to start anticoagulation therapy in patients with MN. This optimal threshold should be discussed.

7. Please correct several typos:
- Abstract: "nephrotic syndrome"
- Page 3 line 120: "tend to be older"
- Page 11 line 535: "obinutuzumab"
- Figure 1 legend: "anergic-apoptotic"

Author Response

1. Could you detail which auto-immune diseases (lupus, IgG4-RD, etc.) and which cancers are the most associated with secondary MN?

Autoimmune diseases such as systemic lupus erythematosus and rheumatoid arthritis [17,18], or malignancy,especially carcinoma [19-21],.  How extensive should be the investigation for detecting an ‘occult’ underlying cancer in a patient with MN is still a matter of debate. While current guidelines prioritize the measurement of serum anti-PLA2R antibodies for diagnosing primary MN, it has been reported that several  secondary MN cases may also test positive for serum anti-PLA2R (see later). At least in heavy smokers and in the elderly patients, however, an extensive investigation is advisable, In these patients and in those with circulating anti thrombospondin 1 domain containing 7 A -antibodies, in addition to a thorough history and physical examination, a chest radiogram (preferably a computed tomography because of its higher sensitivity), a digital rectal exam of the prostate, a thorough breast examination and mammography in women, and renal ultrasonography (usually already done in connection with a kidney biopsy) are suggested. A gastroscopy and a colonoscopy (or at the very least a stool occult blood examination) can also be done in older patients or in the presence of upper gastrointestinal symptoms, early satiety, unexplained weight loss, or iron-deficiency anemia,. 

2. SLGT2 inhibitors should also be mentioned as efficient molecules to decrease proteinuria as symptomatic treatment.

SLGT2I have been mentioned (143)

3. Are high anti-PLA2R levels and the presence of epitope spreading in favour of early immunosuppresive treatment?

Indeed high titers of PLA2R-antibodies are associated with decreased rates of response to treatment while low levels of antibodies are associated with high rate of spontaneous remission.

It has been reported that patients with inner epitopes are more resistant to therapy. (34-40)

4. Please discuss the potential appearance of anti-Ritiximab antibodies and the potential interest of Obinutuzumab in patients with relapse of PLA2R-positive PMN after Rituximab displaying anti-RTX antibodies. 

Anti-drug antibodies (ADA)may develop in Rituximab-treated patients but the long-term implications  of ADA in PMN are poorly inderstood.

5. In your opinion, should malignancy be looked for in all cases of PMN?

At least in heavy smokers and in the elderly patients, however, an extensive investigation is advisable, In these patients and in those with circulating anti thrombospondin 1 domain containing 7 A -antibodies, in addition to a thorough history and physical examination, a chest radiogram (preferably a computed tomography because of its higher sensitivity), a digital rectal exam of the prostate, a thorough breast examination and mammography in women, and renal ultrasonography (usually already done in connection with a kidney biopsy) are suggested. A gastroscopy and a colonoscopy (or at the very least a stool occult blood examination) can also be done in older patients or in the presence of upper gastrointestinal symptoms, early satiety, unexplained weight loss, or iron-deficiency anemia.

6. Given that MN are at higher risk of thrombosis than nephrotic syndrome from other causes, some teams, notably in France, consider a higher threshold of albuminemia (< 25 g/dL) to start anticoagulation therapy in patients with MN. This optimal threshold should be discussed.

We reported that profilactic anticoagulation should be reserved to patients with serum albumin lower than 2-2.5g/dl.

7. Please correct several typos:
- Abstract: "nephrotic syndrome"
- Page 3 line 120: "tend to be older"
- Page 11 line 535: "obinutuzumab"
- Figure 1 legend: "anergic-apoptotic"

Corrected.

Reviewer 3 Report

Comments and Suggestions for Authors

The review is comprehensive and authored by a distinguished colleague who has made significant contributions to the advancement of therapies for primary membranous nephropathy.

Minor Comments:

1.      It would be valuable to include the author’s perspective on the investigation of secondary causes of membranous nephropathy (MN). Historically, there has been considerable debate regarding the necessity of cancer screening prior to initiating immunosuppressive therapy. While current guidelines prioritize the measurement of serum anti-PLA2R antibodies for diagnosing primary MN, it is worth noting that approximately 11% of secondary MN cases may also test positive for serum anti-PLA2R. A discussion of the guidelines in this context, along with the author's views on this issue, would be appreciated.

2.      Given the potential for significant variability in the metabolism of calcineurin inhibitors (CNIs) among different patients, it would be pertinent to assess whether monitoring serum trough levels of these drugs is recommended in such cases. What serum concentrations would be considered optimal in the context of CNI therapy?

3.      There are a few typographical errors that should be addressed.

Author Response

1. It would be valuable to include the author’s perspective on the investigation of secondary causes of membranous nephropathy (MN). Historically, there has been considerable debate regarding the necessity of cancer screening prior to initiating immunosuppressive therapy. While current guidelines prioritize the measurement of serum anti-PLA2R antibodies for diagnosing primary MN, it is worth noting that approximately 11% of secondary MN cases may also test positive for serum anti-PLA2R. A discussion of the guidelines in this context, along with the author's views on this issue, would be appreciated.

As answered to other reviewers:
At least in heavy smokers and in the elderly patients, however, an extensive investigation is advisable, In these patients and in those with circulating anti thrombospondin 1 domain containing 7 A -antibodies, in addition to a thorough history and physical examination, a chest radiogram (preferably a computed tomography because of its higher sensitivity), a digital rectal exam of the prostate, a thorough breast examination and mammography in women, and renal ultrasonography (usually already done in connection with a kidney biopsy) are suggested. A gastroscopy and a colonoscopy (or at the very least a stool occult blood examination) can also be done in older patients or in the presence of upper gastrointestinal symptoms, early satiety, unexplained weight loss, or iron-deficiency anemia.

2. Given the potential for significant variability in the metabolism of calcineurin inhibitors (CNIs) among different patients, it would be pertinent to assess whether monitoring serum trough levels of these drugs is recommended in such cases. What serum concentrations would be considered optimal in the context of CNI therapy?

This writer does not believe in tough levels. Some investigators suggest monitoring calcineurin inhibitors (CNIs) based on trough levels. However, while therapeutic drug monitoring can provide information on serum concentrations, it does not offer insight into intracellular levels. Additionally, various conditions and medications can interfere with cytochrome enzymes and the efflux transporter P-glycoprotein. This may lead to serious errors since elevated CNI levels in the blood may be associated with  low intracellular concentrations and vice versa.

3. There are a few typographical errors that should be addressed.

Corrected.

Round 2

Reviewer 2 Report

Comments and Suggestions for Authors

Thank you for taking my suggestions into accound and for answering my questions satisfactorily. I have no further comments. This review is comprehensive and of high quality.